# Time-Resolved X-ray Observation of Intracellular Crystallized Protein in Living Animal

**DOI:** 10.3390/ijms242316914

**Published:** 2023-11-29

**Authors:** Masahiro Kuramochi, Ibuki Sugawara, Yoichi Shinkai, Kazuhiro Mio, Yuji C. Sasaki

**Affiliations:** 1Graduate School of Science and Engineering, Ibaraki University, Hitachi 316-8511, Japan; 23nm923h@vc.ibaraki.ac.jp; 2Molecular Neurobiology Research Group, Biomedical Research Institute, National Institute of Advance Industrial Science and Technology (AIST), Tsukuba 305-8566, Japan; yoichi-shinkai@aist.go.jp; 3AIST-UTokyo Advanced Operando-Measurement Technology Open Innovation Laboratory (OPERANDO-OIL), National Institute of Advanced Industrial Science and Technology (AIST), Kashiwa 277-8565, Japan; kazu.mio@aist.go.jp; 4Graduate School of Frontier Sciences, The University of Tokyo, Kashiwa 277-8561, Japan; ycsasaki@edu.k.u-tokyo.ac.jp

**Keywords:** crystallized protein, *Caenorhabditis elegans*, diffracted X-ray blinking, in vivo, fluorescent recovery after photobleaching

## Abstract

Understanding the cellular environment as molecular crowding that supports the structure-specific functional expression of biomolecules has recently attracted much attention. Time-resolved X-ray observations have the remarkable capability to capture the structural dynamics of biomolecules with subnanometre precision. Nevertheless, the measurement of the intracellular dynamics within live organisms remains a challenge. Here, we explore the potential of utilizing crystallized proteins that spontaneously form intracellular crystals to investigate their intracellular dynamics via time-resolved X-ray observations. We generated transgenic *Caenorhabditis elegans* specifically expressing the crystallized protein in cells and observed the formation of the protein aggregates within the animal cells. From the toxic-effect observations, the aggregates had minimal toxic effects on living animals. Fluorescence observations showed a significant suppression of the translational diffusion movements in molecules constituting the aggregates. Moreover, X-ray diffraction measurements provided diffraction signals originating from these molecules. We also observed the blinking behaviour of the diffraction spots, indicating the rotational motion of these crystals within the animal cells. A diffracted X-ray blinking (DXB) analysis estimated the rotational motion of the protein crystals on the subnanometre scale. Our results provide a time-resolved X-ray diffraction technique for the monitoring of intracellular dynamics.

## 1. Introduction

Macromolecular crowding (MMC) in the cell has been a topic of recent interest in biology. Approximately 20–30% of the intracellular volume is composed of biopolymers and intracellular organelles [1]. In such a crowded environment, intracellular biomolecules are not capable of the same diffusion movements as in the in vitro environment [2]. Several phenomena have also been reported, in which intermolecular interactions and molecular conformation do not match in vivo and in vitro [3,4,5,6]. The cause of this discrepancy has been suggested to be due to MMC, and attempts to measure the intracellular MMC environment have been vigorously pursued. In MMC environments, rotational diffusion, as well as translational diffusion, is important information for understanding the crowded environment [7,8,9,10]. For example, polarization-dependent FCS (pol-FCS) using a visible light can detect rotational diffusion in cells [11,12]. While techniques for measuring intracellular dynamics are advancing, there is currently a limited range of in vivo measurement methods available for quantitatively assessing the MMC environment within cells. Specifically, there exists a scarcity of techniques designed to measure intracellular dynamics utilizing X-rays.

Diffracted X-ray tracking (DXT), an X-ray measurement technique, is an X-ray diffraction-based molecular dynamics technique that utilizes a pink-beam X-ray to track the X-ray diffraction spots originating from gold nanocrystals attached to proteins [13]. Diffraction spots move in the θ (tilt) and χ (twist) directions on a 2D detector, and this represents the motion of the protein molecule. DXT has been successful in determining the structure-specific dynamics of proteins, such as acetylcholine-binding proteins [14], and TRPV1 channels [15]; these play vital roles in functional expression. However, DXT requires synchrotron radiation facilities and uses high-flux X-rays, which may induce physical and chemical effects on biomolecules due to intense radiation damage in long-term observation. In response to these challenges, we developed the diffracted X-ray blinking (DXB) method, which uses monochromatic X-ray beams. In DXB, diffraction spots entering and exiting the Debye–Scherrer ring, satisfying the Bragg condition, appear to blink. This blinking contains information regarding the tilt and twist motions of the biomolecules. The time-resolved diffraction intensity is analysed using the autocorrelation function (ACF), and the rotational motion is assessed as an exponential relaxation of the ACF curve [16]. DXB enables the observation of molecular motion over extended time scales with low X-ray doses and can be conducted at both laboratories with X-ray sources and synchrotron radiation facilities. DXT and DXB have been successfully used to study the detailed dynamics of various membrane proteins. Nonetheless, the establishment of DXT and DXB techniques for measuring intracellular dynamics has been hindered by the technical challenges associated with introducing nanoparticles into cells.

The spontaneous crystallization of several protein molecules within cells has been reported [17]. Examples include protein storage in plant seeds [18] and polyhedrin coats in virus-infected insect cells [19]. Recently, the crystallizable and photoactivatable (Xpa) molecule was discovered from the stony coral *Favia favus* (Kikume-ishi in Japanese), and Xpa has been reported to exhibit fluorescent properties and to intracellularly crystallize [20]. Considering these characteristics, the Xpa molecule may be a useful probe for measuring intracellular dynamics through X-ray observations. However, it remains uncertain whether this molecule crystallizes within living animal cells or whether it produces distinct X-ray diffraction images when crystallized in vivo. Further experimental validation is required to confirm these aspects.

In this study, we generated transgenic *Caenorhabditis elegans* expressing the crystallized fluorescent protein Xpa in the cells to challenge the development of the intracellular DXB technique. We observed aggregates of the Xpa molecules in cells. These aggregates had minimal effects on the behaviour and physiological functions of living animals. The changes in the fluorescence intensity of the Xpa molecules observed using fluorescence recovery after photobleaching (FRAP) indicated that translational diffusion motion was significantly suppressed when the molecules formed aggregates. X-ray diffraction (XRD) measurements also confirmed the crystalline nature of the Xpa aggregates by providing diffraction signals derived from Xpa. Moreover, we observed the blinking behaviour of the diffraction spots, indicating that these intracellular crystals exhibited rotational motion. A DXB analysis indicated that the Xpa crystal grains had temperature-dependent motion at the subnanometre scale, allowing for the detection of the fluctuating properties of the intracellular crowding environment.

## 2. Results

### 2.1. Observation of Crystallized Protein in the Living Animal Cells

The crystallized protein Xpa H62Q (Xpa) has previously been reported to exhibit the growth of the rhombic-shaped crystals measuring several microns in size upon introduction into the HEK293 cells [20]. Here, we generated transgenic *C. elegans* that specifically express the Xpa gene in the body wall muscles or neuronal cells. Since the Xpa molecule is a fluorescent protein, we easily identified the Xpa-expressing animals under a fluorescence microscope (Figure 1A). Interestingly, the Xpa molecule displayed puncta-like aggregation. Using super-resolution microscopy, we further examined the shape of the Xpa puncta, determining the presence of the rhomboidal aggregates (Figure 1B). These aggregates were observable in both the neuronal and muscle cells (Figure 1C,D). In both the body wall muscle and neuronal cells, aggregates on the order of a few microns were consistently formed. We calculated the surface area of these aggregates based on the image data, resulting in a median puncta area of 1522 µm^2^ in the body wall muscle and 646 µm^2^ in the neurons. The size of the Xpa puncta formed in the body wall muscle was significantly larger than that formed in neurons (Figure 1E).

To further characterize the Xpa molecular aggregates, we examined their shape, aggregation sites, and the number of aggregates per animal. Aggregate shapes were classified into “rhombic”, “rectangular”, and other polygonal shapes (Appendix A). Approximately 60% of Xpa molecules exhibited a “rhombic” shape in both muscle and nerve cells (Appendix A). To determine the body region with the highest abundance of the Xpa aggregates, we examined the number of aggregates by distinguishing three expression sites: head, abdomen, and tail (Appendix A). The head region had the highest percentage of aggregates, accounting for more than 60% of the total, in both the body wall muscles and neurons (Appendix A). Additionally, we investigated the number of the Xpa molecular aggregates formed per animal. The distribution of the number of aggregates per animal is shown in Appendix A. Approximately two aggregates were formed per animal, with no significant differences between the body wall muscles and neurons. No differences in the number of aggregates formed depending on the site of expression were observed.

### 2.2. Toxicity Effect of the Crystallized Protein In Vivo

To investigate the effect of the Xpa aggregates on cellular and physiological functions, we evaluated the number of eggs laid, head thrashing behaviour, and lifespan of animals. First, we assessed the egg-laying function by counting the total number of eggs laid per individual animal. The average numbers of eggs were 93 and 115 in the Xpa-expressing animals in the body wall muscles and in the neurons, respectively. In contrast, the wild-type animals laid a total of 101 eggs (Figure 2A). The egg-laying activity of the Xpa-expressing animals did not significantly differ from that of the wild-type animals. Therefore, Xpa aggregates had little direct impact on egg laying. Next, we investigated the effects of the Xpa aggregates on animal locomotion. *C. elegans* regulates locomotion by repeatedly contracting and relaxing the body wall muscles. Abnormalities in muscle function affect the thrashing response. Here, we assessed head thrashing behaviour to observe whether the Xpa aggregates in the body wall muscles have any toxic effects on muscle function. The average number of head thrashings per minute was 128 for the wild-type animals, 123 for the Xpa-expressing animals in the body wall muscles, and 125 for the Xpa-expressing animals in the neurons. Based on this, the Xpa aggregates did not appear to interfere with the muscle function itself or with muscle control via neural function (Figure 2B). In the lifespan assay, we compared the survival of animals over time among the following groups: wild type and the Xpa expression in their muscles and in their neurons. The differences between these groups were very minimal (Figure 2C). These results indicated that the aggregates were unlikely to have toxic effects on the lifespan. An analysis of the number of eggs laid, head thrashing, and lifespan indicated that this small number of Xpa aggregates was unlikely to be toxic to animals.

### 2.3. Translational Diffusion of Molecules in Aggregates Is Considerably Suppressed

As shown in Figure 1A–D, we observed the spontaneous aggregation of the Xpa molecules within the cells in a living animal. However, the dynamic properties of these aggregates were not clear. To investigate the dynamic property, we used the fluorescence recovery after photobleaching (FRAP) method [21,22], which evaluates the translational diffusion movement of molecules. As the Xpa molecule itself is a fluorescent protein, FRAP exploits its fluorescent properties. In FRAP, a powerful excitation laser beam locally photobleaches the fluorescence of the Xpa molecules within the cell, and the subsequent recovery of fluorescence intensity resulting from the entry of fluorescent molecules into the irradiated area is measured and analysed (Figure 3A). The fast fluorescence recovery indicates rapid translational diffusion, while a slow recovery indicates slower diffusion. When nonaggregated Xpa molecules in both the muscles and neurons were irradiated with a powerful laser, a temporary loss of fluorescence was observed in the irradiated area, and this fluorescence was rapidly recovered (Figure 3B,C below). This result indicated that the nonaggregated Xpa molecules, both within and surrounding the irradiated region, quickly moved to the irradiated area through free diffusion, leading to the recovery of fluorescence intensity. In contrast, the aggregated Xpa molecules lost their fluorescence upon laser irradiation near the centre of the aggregates, and their fluorescence was not recovered over time (Figure 3B,C top). These molecules could not freely diffuse, indicating that their movement was likely constrained by the structure formed by the Xpa molecules themselves (Appendix A).

### 2.4. X-ray Blinking of the Crystallized Protein in Living Animal Cells

FRAP has shown that the translational diffusion of the Xpa molecules forming aggregates was severely constrained. We hypothesised that this constrained molecular movement potentially occurred because the Xpa aggregates adopted a regularly arranged crystal structure, which was extremely rigid. To investigate whether the Xpa molecular aggregates were crystalline, X-ray diffraction (XRD) images were captured within semi-immobilized animals expressing Xpa in the body wall muscles (Figure 4A). In the wild-type animals, no diffraction spots or rings were observed, as shown in Figure 4B. Conversely, Xpa-derived diffraction signals were obtained from the Xpa-expressing animals. Interestingly, a blinking diffraction spot was observed in the time-resolved XRD images (Figure 4C). The blinking phenomenon was reported in several previous studies to be closely associated with the rotational motion of crystal particles [16,23]. Due to the rotational motion of the Xpa crystals, diffraction spots moved in and out of the Debye–Scherrer ring. These diffraction spots only appeared when the Bragg condition was satisfied, resulting in the appearance of blinking. This blinking behaviour indicated that the Xpa crystal grains within cells rotated over time.

### 2.5. Diffracted X-ray Blinking of the Crystallized Proteins In Vivo

Here, we utilized the blinking phenomenon of X-ray diffraction spots originating from the Xpa crystal to assess the dynamic properties of the crystal. In diffracted X-ray blinking (DXB), the rotational motion of crystals is evaluated as an ACF decay constant by analysing the time variation of the X-ray diffraction intensity as an autocorrelation (Appendix A). In conventional DXB measurements, the rotational motion of protein molecules is assessed through the chemical labelling of the target protein molecule with gold nanocrystals and by analysing the diffraction spots from these nanocrystals. The Xpa molecule itself is crystalline and produces clear diffraction spots, enabling label-free measurements without gold nanocrystals. An ACF analysis was conducted using 2000 XRD images, each with an exposure time of 50 ms. As shown in Figure 5A, the ACF curves exhibited decay over lag time, indicating that the Xpa crystals rotated. Remarkably, the ACF decay constant of the Xpa crystals appeared to increase in a temperature-dependent manner (Figure 5B,C). This result implied that higher temperatures enhanced the rotational motion of the crystals within the cells in the living animal. The estimated rotational diffusion at 50 °C ranged from 4.677 × 10^−4^ to 1.170 × 10^−3^ nm^2^/s. The DXB analysis showed that Xpa grains have temperature-dependent motion on the subnanometer scale. An intracellular DXB using Xpa crystals can detect dynamical changes in an MMC environment at the sub-nanometer scale. We established a label-free in vivo DXB measurement using Xpa crystals expressed in the cells.

## 3. Discussion

In this study, we generated transgenic animals that expressed the crystal and fluorescent protein Xpa molecules. An aggregation of the Xpa molecules was observed in the living animal cells, with the size of the Xpa aggregates being larger when expressed in muscles than in neurons. The toxic effects of these aggregates were assessed in terms of the number of eggs laid, the number of head thrashings, and lifespan. The results indicated that there was little effect of the aggregates. The fluorescence intensity changes in the Xpa molecules observed through FRAP indicated that the translational diffusion movements of the molecules were significantly suppressed when they formed aggregates. This result was consistent in both muscles and neurons. Additionally, XRD measurements enabled the acquisition of the Xpa-derived diffraction signals, confirming that the Xpa aggregates were crystalline. The crystals were thin enough to obtain X-ray diffraction. We also observed the blinking behaviour of the diffraction spots, indicating that these crystals exhibited rotational motion within the living animal cells. From the DXB analysis, we estimated the rotational motion of the Xpa crystals to be on the subnanometre scale. This motion was also found to increase in a temperature-dependent manner.

With regard to the crystallised molecule used in this study, it has been found in previous studies to crystallise in multiple species of cultured cells. In this study, we found that this protein molecule crystallises in *C. elegans* cells. It is not known whether Xpa crystallises in cells of other organisms, but the genes of *C. elegans* are similar to those of mice and humans. It is thought that there are no major differences in the constituent molecules in the cells. Hence, it is likely that they will crystallise in mammals such as mice. In addition, various organisms have been reported to form crystallites in the cells [17], and gene transfer or similar would not be necessary in these organisms. Certain medical conditions are linked to crystal precipitation, such as urinary tract stones [24], gallstones [25], and gout [26]. While their application to humans at this stage poses challenges, there is potential to explore these conditions as potential areas of application.

No significant differences were observed between the Xpa-expressing and wild-type animals in the three toxicity assessments of egg-laying, thrashing, and life span; these results indicated that the Xpa crystals were unlikely to have toxic effects on animals. Importantly, crystallized proteins are not biotoxic and can, therefore, be effectively used for label-free DXB to obtain intracellular dynamics without the use of nanocrystals. On the other hand, toxic effects have not been verified at the individual-cell level. An assessment of activity at the single-cell level through calcium imaging would be useful [27,28]. 

As shown in the FRAP data in Figure 3, translational motion within the Xpa crystals was not observed on the micrometre scale. This result was potentially caused by the rigid crystal structure with a regular arrangement retained by the Xpa crystals. Rotational motion may also be suppressed, but since DXB detects rotational motion with subnanometer precision, it was probably able to detect the rotation of the crystal grains on the microscopic spatial scale. To more thoroughly investigate this rotational motion, diffracted X-ray tracking (DXT) measurements using pink beams were used. DXT enables the visualization of the rotational motion of crystal molecules in two directions, θ and χ [14,15]. A further observation using DXT is essential for a comprehensive understanding of the dynamic properties of crystallized proteins.

The X-ray diffraction signal from the crystallized protein within the cells in the living animal was relatively clear, indicating that it could be used effectively as a technique for the measurement of the intracellular crowding environment. Our results are highly significant for the implementation of the measurement of intracellular MMC dynamics. However, some concerns and technical improvements will need to be implemented in the future to address them. 

The measured intracellular dynamics will require a careful evaluation as they are influenced by crystal size, shape, and MMC environment. With regard to crystal size, previous DXT measurements have reported a reasonable correlation between peak angular velocity values and nanocrystal size (Ref.). The larger nanocrystals mean that the angular velocity is reduced. In addition to the size of the crystals, rotational diffusion is also affected by the shape of crystals [29,30] and the surrounding environment. For example, in silver halide fabricated through epitaxial growth, the decay constant of the annealed silver bromide was higher than that of the non-annealed silver chloride, even though they were approximately of the same size. The decay constant did not show a simple correlation with the size of the crystal grains. An atomic force microscopy (AFM) shape analysis indicated that non-annealed silver chloride exhibited a flatter shape compared to annealed silver bromide. The density of the crystalline particles and the gaps between the particles on the thin film also differed between silver bromide and silver chloride [23]. These multiple factors for shape and environment may limit the rotational motion. It was reported that the Xpa crystals formed rhombic crystals in HEK293 cells and rod-shaped crystals in hippocampal neurons [20]. In *C. elegans* cells, rhomboidal crystals were more common, with the presence of rectangular and polygonal-shaped crystals. Notably, the behaviours of crystal grains could vary due to these differences in shape. Since the crystals could show different structure-specific rotations, approaches such as DXB measurements are needed along with the evaluation of the crystal shape in visible light. We have already introduced microscopes that allow for fluorescence observation during X-ray measurements, enabling these approaches to be tested in practice. 

In addition, crystals of various sizes ranging from a few micrometers to several tens of micrometers were formed in the cell. Further improvements are needed to accurately detect the dynamics of the local space within the cell, such as the control of Xpa expression level and introduction of X-ray focusing techniques.

Intracellular DXB can effectively detect subnanometer crystal rotations, but it cannot discern micrometer-level changes. For such macroscopic behaviours, visible-light observation data such as pol-FCS would be useful [12]. The integration of visible light observations with X-ray measurement techniques will capture a variety of microscopic to macroscopic intracellular dynamics and will provide new insights into in vivo phenomena such as MMC environments, molecular interactions, and protein folding.

## 4. Materials and Methods

### 4.1. Strains and Cultivation

All animals were cultivated on standard nematode growth media (NGM) agar plates seeded with *E. coli* OP50 at room temperature (~22 °C). We used the following strains: wild-type *C. elegans* variety Bristol strain (N2), CMS50 *Ex[H20p::Xpa H62Q]*, and CMS67 *Ex[myo-3p::Xpa H62Q]*.

### 4.2. Molecular Biology and Transgenic Animals

Standard methods for molecular biology were used to construct plasmid DNAs. Since Xpa is photoactivated (photoconvertible) from green to red using 405 nm light irradiation, we generated transgenic animals expressing the Xpa H62Q variant (nonphotoconvertible) in a cell-specific manner. Xpa H62Q was provided by RIKEN BRC through the National BioResource Project of the MEXT/AMED, Japan. For the expression of the genetically encoded Xpa H62Q (Xpa) protein, the coding sequence was inserted between the BamHI and EcoRI sites of the pPD95.77 vector. Then, the promoter region for cell-specific expression of the cDNAs was inserted between the XbaI and ApaI sites for *myo-3p* and between the BamHI and ApaI sites for *H20p* in the resulting pPD95.77/Xpa plasmid. We used the myo-3 promoter for cell-specific expression of body wall muscles and the H20 promoter for cell-specific expression of neuronal cells.

To generate transgenic animals, plasmid DNAs were injected into N2 (Bristol) wild-type animals using a standard microinjection method [31].

### 4.3. Microscopic Observations and FRAP 

Transgenic adult animals were immobilized on a 2% agarose pad containing 50 mM sodium azide in the M9 solution for anaesthesia. Differential interference contrast images and fluorescence images were observed using an inverted fluorescence microscope (Olympus IX51, Tokyo, Japan) and captured using an ORCA-Flash 4.0 CCD camera (Hamamatsu Photonics, Hamamatsu, Japan) controlled using HCImage software (https://hcimage.com/) (Hamamatsu Photonics, Hamamatsu, Japan). Super-resolution images were obtained using the CSUW1-SoRa super-resolution unit (Yokogawa Electric Corp., Tokyo, Japan). Image analysis was performed using ImageJ/Fiji software version 2.9.

The size and shape of the protein aggregates in the cells were evaluated using fluorescence images. The Xpa aggregate shapes were categorized as rhombic, rectangular, or polygonal based on the acquired fluorescence images. To determine the aggregate size, the area of the crystals was calculated using ImageJ/Fiji software version 2.9.

The fluorescence recovery after photobleaching (FRAP) method was used to investigate the molecular translational diffusion within the aggregates formed in the cells. Since the Xpa molecule itself is a fluorescent protein, the recovery of the Xpa fluorescence after photobleaching was monitored. Transgenic adult animals were immobilized on a 2% agarose pad in the M9 solution with the addition of 50 mM sodium azide. FRAP experiments were conducted using an FV1000 confocal laser scanning microscope (Olympus) with an exposure time of 200 ms. The resulting images were analysed using ImageJ.

### 4.4. Toxicity Assessment

In the egg-laying count experiment, a single adult animal was placed on a freshly seeded plate and allowed to lay eggs for 24 h, and the eggs were counted. The animal was subsequently transferred to a new NGM plate, and this process was repeated every 24 h for a total of 3 days. *n* ≥ 13. Student’s *t* test was performed.

Head thrashing was assessed by placing a single adult animal in an M9 buffer solution and counting the number of thrashings per minute. n ≥ 15. Student’s *t* test was performed.

Lifespan was assessed by placing over 20 larvae on a plate with food and recording the number of dead animals daily. Animals were considered dead if they did not respond mechanically after three light touches to the head with a platinum wire. n ≥ 20 (group = 5).

### 4.5. Sample Preparation for X-ray Diffraction Measurement

One millilitre of the M9 buffer was added to the NGM plate on which the animals were cultured, and the turbid fluid containing the animals was collected in a 1.5 mL tube. The tubes were then centrifuged at 3000 rpm for 3 min, and the supernatant fluid was discarded. The turbid fluid at the bottom of the tube, containing several hundred animals, was added dropwise onto a polyimide film. Approximately 50 µL of 50 mM sodium azide was added to semi-fix the animals. After removing the excess water with a paper towel, the polyimide film was placed over the sample, and the sample was set in the sample holder.

### 4.6. Diffracted X-ray Blinking

DXB was conducted using laboratory X-ray source (MicroMax-007 HF, Rigaku: Cu anode, wavelength (λ) = 1.54 Å, 40 kV, 30 mA). Time-resolved diffraction images were recorded using a 2D photon-counting detector (Pilatus 200K-A, Dectris, Baden, Switzerland). The sample-to-detector distances were 30 mm. Under temperature control, XRD images were continuously acquired for a total of 2000 frames. The exposure time per frame was set to 50.0 ms. The X-ray diffraction intensities from the crystallized protein molecules were analysed using the following autocorrelation function (ACF):$$I\left(\tau \right)=\frac{\left\langle I\left(t\right)I\left(t+\tau \right)\right\rangle }{\left\langle I{\left(t\right)}^{2}\right\rangle }$$
where *I(t)* represents the diffraction intensity, the brackets < > indicate the time-averaged value, and τ represents the lag time. The ACF curves were fitted to an exponential curve using $\mathrm{ACF}\left(\tau \right)=A*\exp\left(-\Gamma t\right)+y$, where *A* represents the amplitude, *y* is the conversion factor, and Г is the decay constant. We selected decay constants to satisfy the following conditions: (I) 0 < *y*, 0 < *A*, and 0 < *Г*, and the (II) residual values between the fitted and actual ACF curves were less than 1.0 [15,18]. These calculations were applied to all pixels. The distribution of the decay constants of the Xpa diffraction ring was visualized using histograms and box plots to estimate the dynamic behaviour of the protein molecules.

The rotational diffusion coefficient *D_R_* was calculated from the DXB data using methods from previous studies [19,32,33]. The ACF decay constant Г for a single pixel is directly related to the rotational diffusion coefficient of a single particle and is expressed by the following equation:$$D_{R}=\frac{\varphi_{\theta }^{2}\Gamma }{4}.$$

Here, *φ_θ_* represents the angular displacement. Note that calculations were based on the assumption that diffraction spots exist over an angular range wider than a single pixel. The calculation of *φ_θ_* was performed as follows: the size of a single pixel on the detector was 172 µm × 172 µm. The distance from the centre of the diffraction ring to the Xpa diffraction signal was 15.996 mm on the low-angle side and 16.512 mm on the high-angle side. A camera length of 30 mm was used, and the calculation was performed using the arctan function, resulting in values of 28.067° on the low-angle side and 28.828° on the high-angle side. For this calculation, the median value of the ACF decay constant at 50 °C was used (Appendix A).

## Figures and Tables

**Figure 1 ijms-24-16914-f001:**
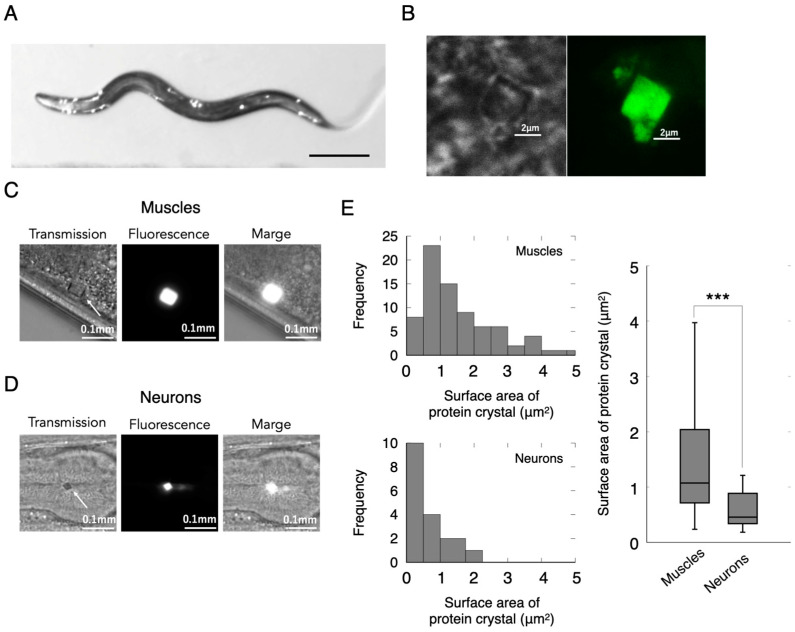
Aggregated proteins within the cells of a living animal. (**A**) Microscopy images of Xpa-expressing animal. Scale bar 0.2 mm. (**B**) Super-resolution imaging of Xpa in the animal cell. The shape of the Xpa protein in the muscle cells (**C**) and neurons (**D**) of animals. (**E**) Distribution of the surface area of the Xpa protein. *** *p* < 0.001.

**Figure 2 ijms-24-16914-f002:**
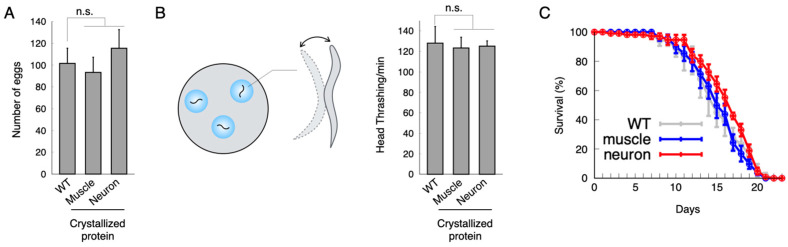
In vivo toxicity effects of the aggregated protein. (**A**) Number of eggs laid. (**B**) Number of head thrashings in the M9 buffer environment. Student’s *t* test was performed. “n.s.” indicates no significance. (**C**) Lifespan analysis.

**Figure 3 ijms-24-16914-f003:**
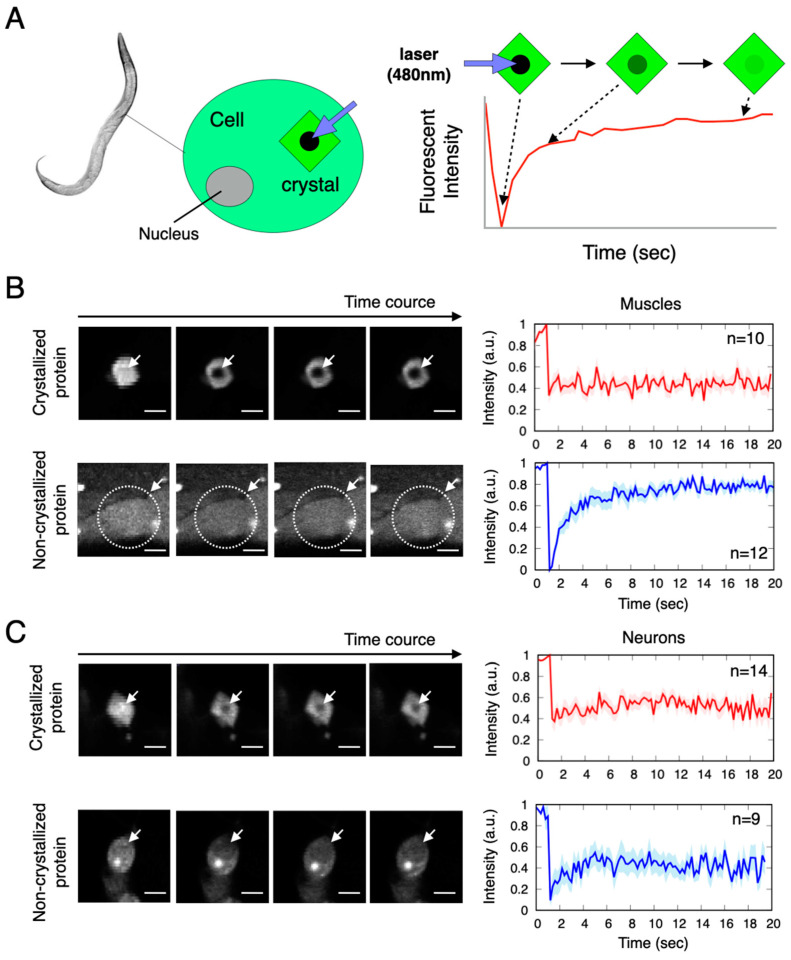
Translational diffusion motion of the Xpa molecules within aggregates. (**A**) Schematic of the FRAP experiments of the Xpa aggregates in living animal cells. FRAP experiments of the Xpa-expressing animal in the body wall muscle (**B**) and neurons (**C**). Fluorescence images of the time course during the FRAP experiments (**left**) and the time profile of fluorescence intensity (**right**). The arrows in the image represent photobleached areas by laser irradiation. Scar bar 0.05 mm.

**Figure 4 ijms-24-16914-f004:**
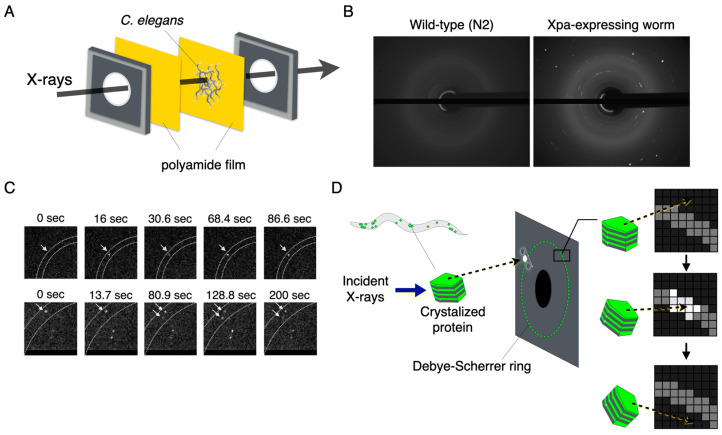
X-ray blinking phenomena of the crystallized protein in the animal cells. (**A**) XRD setup for the in vivo observation. (**B**) XRD images of the wild-type animal and Xpa-expressing animals. (**C**) X-ray blinking phenomena derived from the Xpa-expressing animals. These arrows indicate the diffraction spots. (**D**) Schematic illustration of X-ray blinking inducing the rotational motion of the crystallized protein.

**Figure 5 ijms-24-16914-f005:**
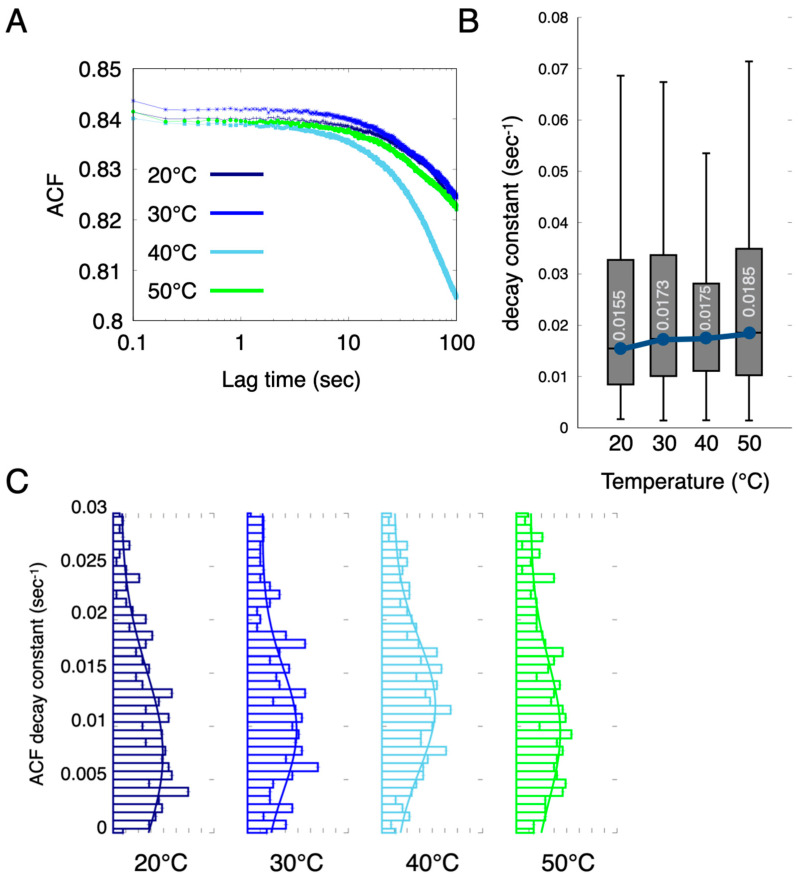
Diffracted X-ray blinking (DXB) observation of the intracellular crystallized protein in a living animal. (**A**) ACF curves from the temporal profile of the diffraction intensity in the Xpa crystals in the living animal cells. (**B**) Boxplot of the ACF decay constants. The boxes show the median and first and third quartiles. The inset number shows the median value of the decay constant. (**C**) Distribution of the ACF decay constants at 20−50 °C.

## Data Availability

The data that support the findings of this study are available from the corresponding author upon reasonable request.

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
