# Peer review of "Time-Resolved X-ray Observation of Intracellular Crystallized Protein in Living Animal"

_ijms, 2023, doi:10.3390/ijms242316914_

Round 1

Reviewer 1 Report

Comments and Suggestions for Authors

Kuramoochi et al. used the time-resolved X-ray diffraction technique to study translational and rotational motions of intracellular crystals spontaneously formed in living cells. They first showed that the crystal had minimal toxic effects on the animal. Then, they observed suppressed translational diffusion of molecules in aggregates, but relatively unobstructed rotational motion of the crystals. I think the manuscript has successfully demonstrated the effectiveness of an innovative and groundbreaking experimental technique that allows for nanometer-level measurements of molecular dynamics in living cells. The texts are well-written, and results are presented clearly. Therefore, I believe the manuscript can be published as is. I have only one comment:

1.     Although translational movements were not observed for aggregated crystals, the authors can still provide methods for calculating translational diffusion coefficients using their technique for future studies.

Reviewer 2 Report

Comments and Suggestions for Authors

This seems like a nice introduction of a new method for studying the intracellular environment.

There are two new elements compared with the earlier work: the crystals are those of the protiens being studied, rather than

exogenous nanoparticles, and a monochromatic beam is used, making the experiment an instance of XPCS.

I wonder how broad the application of this method can be.  In present form, it requires transgenic organisms which can express

protiens (not necessarily fluorescent) which aggregate to form crystals.  Further, the tissue being looked at has to be thin

enough for the signal from a few crystals to show up against the background of scattering from the tissue.

Curious that the abstract doesn't specify the sort of animal involved.  Only at pg. 2, line 78 do we learn that it's C. elegans.

L47 DXT is not a quantum technique.  There's nothing particularly quantum about XRD.  Ref 13 has nothing to do with proteins.

The nanoparticles were embedded in agarose gel.  The "crystals" involved are Mo-Si multilayers, not Au nanoparticles.  I suspect

that the authors have two references confused.  Given that X-ray beams are not specifically quantum probes, why are they included

in a special issue with 'quantum' in the name?  There is altogether too much loose use of the word, both in science and in the

popular press.

L368 This assumes that the diffraction spot exists over a wider angular range than a single pixel.  Is that correct?  Is it

limited by the angular divergence of the incident beam or the mosaicity of the aggregate?  Important details relevant to

reproducting the work, such as the X-ray wavelength, are not given.  The details of the 'physics' methods should be discussed

in as much detail as those of the 'biology' methods.

Reviewer 3 Report

Comments and Suggestions for Authors

The manuscript submitted by M. Kuramochi describes the formation of crystal of protein Xpa in cells of a nematode, the analysis of their toxicity, and their dynamics. It was observed that these crystals are not toxic, they do not translate, and they rotate. Per se, all these observations are little interesting but they are good preliminary works on the analysis of in vivo crystals and in vivo properties.

The manuscript is perhaps not very well written and appears confuse time to time. However, it is an interesting piece of science that should be read by other scientists.

I have two main concerns.

1) Lines 167-169. The fact that molecules do not translate does not depend on the fact that they are crystallized. This depends on the fact that crystals do not translate. Perhaps they are too heavy.

2) Crystal dynamics must be strongly dependent on crystal shape and size. This is mention only very superficially in the manuscript. However, it should be better discussed and one is expected to read some hypothesis on how shape and size might be controlled.

Minor

3) The sentence “approximately two aggregates… “ of line 117 is repeated in line 141.

4) Figures 4b and 4c. Time cource and Frame are inappropriate. One should see here real units (seconds, minutes, …).

5) Figure 5A. It is curious that line 50 degrees is in between line 40 degrees, on the one hand, and lines 20 and 30 degrees, on the other hand.
